# Preschool Environmental Factors, Parental Socioeconomic Status, and Children’s Sedentary Time: An Examination of Cross-Level Interactions

**DOI:** 10.3390/ijerph16010046

**Published:** 2018-12-25

**Authors:** Suvi Määttä, Hanna Konttinen, Reetta Lehto, Ari Haukkala, Maijaliisa Erkkola, Eva Roos

**Affiliations:** 1Folkhälsan Research Center, Topeliuksenkatu 20, 00250 Helsinki, Finland; reetta.lehto@folkhalsan.fi (R.L.); eva.roos@folkhalsan.fi (E.R.); 2Faculty of Social Sciences, University of Helsinki, 00014 Helsinki, Finland; hanna.konttinen@helsinki.fi (H.K.); ari.haukkala@helsinki.fi (A.H.); 3Department of Food and Nutrition, University of Helsinki, 000014 Helsinki, Finland; maijaliisa.erkkola@helsinki.fi; 4Department of Public Health, Clinicum, University of Helsinki, 00014 Helsinki, Finland

**Keywords:** children, sedentary lifestyle, preschool, socioecological model, socioeconomic status

## Abstract

Preschool children’s high levels of sedentary time (ST) is a public health concern. As preschool reaches a large population of children from different socioeconomic status (SES) backgrounds, more knowledge on how the preschool setting is associated with children’s ST is relevant. Our aims were to examine (1) the associations of preschool setting (covering social, physical, and organizational level) with children’s ST, and (2) the moderating role of the setting on the association between parental SES and children’s ST. In the cross-sectional DAGIS (increased health and wellbeing in preschools) study, the participating children (*n* = 864, aged 3–6 years) were asked to wear an accelerometer for one week. In total, 779 children had valid ST accelerometer data during preschool hours. Preschool setting and parental SES was assessed by questionnaires and observation. Multilevel linear regression models with cross-level interactions were applied to examine the associations. Early educators’ practice of breaking children’s ST often, more frequent physical activity (PA) theme weeks, and higher number of physical education (PE) lessons were associated with lower children’s ST. Higher parental SES was associated with higher children’s ST in preschools (1) with organized sedentary behavior theme weeks, (2) with a lower number of PA theme weeks, and (3) with a lower number of PE lessons. The factors identified in this study could be targeted in future interventions.

## 1. Introduction

Having an optimal amount of physical activity (PA) and minimized sedentary time (ST) during preschool age (aged 3–5 years) are the key elements in shaping children’s healthy lifestyle habits [1]. The health benefits of regular PA on social, physical, and psychological health are highlighted in multiple research findings [2]. ST is defined as consisting of multiple types of sedentary behaviors (SB) that are characterized by an energy expenditure of less than 1.5 metabolic equivalents while in a sitting, reclining, or lying posture [3]. The health consequences of overall ST, measured by objective measurements (e.g., accelerometers), at this age group are still not well known, but multiple studies using screen time as an indicator of ST have shown negative health consequences on social, physical, and psychological health [4,5]. Current recommendations in several countries are to minimize preschool children’s prolonged ST to one hour and break children’s ST often [6,7,8]. However, recent studies indicate that many preschool children are sedentary for most of their waking hours [9,10]. Although PA and ST have distinct correlates [11], identifying correlates of ST may help in the development of intervention strategies aimed at shifting children’s behavior from sedentary to more active pursuits [12,13].

According to socioecological models [14,15,16], it has been suggested that behavioral correlates are setting-specific [15]. Given that enrollment in preschools has become the norm for the majority of children [17], recognizing correlates of children’s ST in the preschool setting may be valuable. Following the principles of socioecological models, preschool as a setting with structured daily schedules may control children’s ST by either promoting, discouraging, prohibiting, or sometimes demanding children’s activity by giving children only a limited option for deciding on their own behavior [18,19]. Some previous studies have also suggested that children attending preschools have more ST than children attending mainly home-based childcare or preprimary education [20,21,22]. Previous studies have also found that preschool settings can explain the variance in children’s behavior after individual-level factors are taken into account [20,23,24,25]. Therefore, if the aim is to modify behavior in a preschool setting, the focus should be on environmental factors in this setting [26].

Of the preschool setting, physical (what is available, e.g., PA equipment), social (e.g., the attitudes and practices of early educators), and organizational (e.g., regulations related to ST) factors can be separated as distinct environmental factors [27,28]. According to recent reviews [29,30], the physical environmental factors are the most studied, and consistent evidence exists that high-quality outdoor physical environmental factors (e.g., outdoor environment, PA equipment) have an association with reduced children’s ST. There is a lack of consistent evidence on the associations between social and organizational environmental factors and children’s ST, although social environmental factors, such as educator practices and role modeling, are often mentioned as strategies to reduce ST [29,30]. It is therefore not surprising that many of the preschool interventions have had a small-to-moderate effect on decreasing children’s ST, because the key factors to focus in the development of intervention strategies are still poorly known [31,32].

Many studies, including our previous study, have failed to find any socioeconomic status (SES) differences in children’s objectively measured ST [33,34]. Still, researchers have recently been encouraged to continue the analyses by conducting moderation analyses to understand whether a certain setting (e.g., preschool) alters/modifies the associations between SES and children’s behavior [35,36,37]. These recommendations follow the principles of socioecological models, that is, supportive proximal factors—such as the preschool setting—may buffer the influence of more distal processes (e.g., SES) on children’s behavior [14,38]. For instance, low parental SES as a distal factor may have less influence on children’s ST during preschool hours if there is a high-quality preschool setting in terms of reducing children’s ST (e.g., high amount of PA in daily schedules throughout the week, and equipment encouraging PA). Similarly, children from high SES families may have a high amount of ST during preschool hours if there is no support for reducing ST in the preschool setting. [38] To the very best of our knowledge, there are no previous studies examining if the factors in a preschool setting modify the associations between SES and preschool children’s ST. Although various numbers of studies have examined the associations between multiple levels of socioecological models (including, e.g., social and physical preschool environmental factors, indicators of SES) and preschool children’s ST [39,40,41], interactions between the behavioral correlates, the core of the socioecological viewpoint, are ignored [28,42] Studying these interactions may provide insights that aid to tailor preschool intervention strategies appropriately.

We set two aims for our study: (a) To investigate the associations between a set of physical, social, and organizational environmental factors in a preschool setting and preschool children’s objectively measured ST during preschool hours, and (b) to study the moderating role of the preschool setting on the association between parental SES and children’s ST during preschool hours.

## 2. Materials and Methods

### 2.1. Study Context

In Finland, municipalities are responsible for organizing preschool education for children. All children have a right for a preschool place for at least 20 h a week [43]. Approximately 80% of children between the ages of 3 and 5 from different socioeconomic backgrounds attend preschool. Preschool children are enrolled in formal childcare for an average of 30 h or more per week. [44] Preschool care in Finland is subsidized, with a maximum fee of €290 per month to be paid by the wealthiest parents (as of 2017). The family income and the size of the family are taken into account when the fee is determined. The Finnish preschool model is based on learning by playing, and compulsory preprimary education in preparation for official schooling starts at the age of 6 [43].

### 2.2. Sample

The DAGIS (increased health and wellbeing in preschools) cross-sectional study was conducted in eight municipalities around Southern and Western Finland between September 2015 and April 2016. The more detailed information on the DAGIS study protocol can be read elsewhere [45]. Preschools were randomly invited to participate in the study. The number of invited preschools was based on power and sample size calculations [45]. The main recruitment criterion for the preschools was that there needed to be at least one preschool group with children aged 3–6 years. Purely preprimary education groups for 6-year-olds were not included in the study sample. Eighty-six preschools (51% of those invited) in these municipalities gave permission for the study to be conducted in their preschools. Children and families were recruited through preschools. A total of 983 parents (27% of contacted parents) gave written permission for their child to participate in the study. We also expected that at least 30% of children in at least one preschool group in the preschool would participate in the study so that study procedures were implemented. Therefore, 91 parents had a child in preschool with less than 30% of consenting rate. In addition, 28 children had no data that could be used. In total, 864 (24% of invited) children from 66 preschools (39% of invited preschools) participated in the study. These preschools had 159 preschool groups with children aged 3–6 years. Figure 1 shows the flow of the participating preschools and the participants. The University of Helsinki Ethical Review Board in the Humanities and Social and Behavioral Sciences approved the study procedures (6/2015, approved on 25 February 2015).

### 2.3. Measures

#### 2.3.1. Children’s Sedentary Time

Children’s ST was measured using an Actigraph W-GT3X accelerometer (Actigraph, LLC, Fort Walton Beach, Florida). Children wore the accelerometer on their hip 24 h a day for seven days. Actigraph has been validated and used extensively as an objective measure of PA and ST for different age groups and in different contexts [46,47,48]. After data collection, the epoch length was set at 15 s. Periods of 10 min or more at zero accelerometer counts were considered to be nonwear times and were excluded [49]. The possible nap-times were not excluded. For the analyses, the Evenson ST cut-point (≤100 counts per min) was applied [50], having been shown to be a good estimate of free-living ST [51,52].

For the purposes of this study, only the accelerometer data from the preschool time were used. Parent-provided information about daily preschool hours were applied. We set the following wear-time criteria to be included in the analyses: Children needed to be at the preschool for at least 240 min per day for at least two days. Because preschool hours varied between children, the variable was adjusted for the preschool wearing hours. The measure used in this study indicates, therefore, the children’s ST minutes in one hour in preschool. 

#### 2.3.2. Preschool Setting

##### Physical Environmental Factors

Research assistants observed the preschool environmental factors using a comprehensive observation instrument purposefully designed for this study and suitable for the Finnish context. The observation tool included items of the environment and policy assessment and observation instrument (EPAO) [53], items of the national investigation about the PA conditions in Finnish preschools [54], and some additional items developed to meet the aims of this study. Two assistants observed preschools independently and simultaneously. After each observation, the research assistants discussed their answers to reach consensus on the findings. In this study, three measures were derived from the DAGIS observation tool.

Availability of indoor PA equipment is a composite score of two equipment categories, that is, the portable equipment in the group facilities (*n* = 10 observed equipment) and the fixed equipment inside (*n* = 5). The equipment in the group facilities was observed by answering the following options: Yes, in view; yes, in the closet; and no. These answer options were recoded so that the first two answer options were combined to illustrate that the preschool group had certain equipment (1). The fixed equipment was observed so that preschool either had it (1) or not (0). The EPAO-scoring procedures were followed when forming the adjusted score [55]. The adjusted score was formed so that the items in each category were summed up and divided by the number of items and multiplied by ten. After this, these categories were summed up to the adjusted indoor PA equipment measure.

Availability of outdoor PA equipment is a composite score of two equipment categories that is fixed equipment outdoors (*n* = 9) and portable equipment outdoors (*n* = 8). The answer options in both categories were either yes (1) or no (0). The adjusted score was formed in a similar manner as mentioned with respect to the indoor equipment measure. 

Availability of screens is a composite score of all observed screens in preschools. Television, tablet computers, game consoles/DVD/video players, and computer were the observed screens at the preschool. The screens were observed by choosing the right options in the following: In group facilities, in common facilities, and none at all. These answer options were recoded so that the availability either in the group facilities or in the common facilities were combined into one option. After this, all the screen items were summed into one measure. This measure was dichotomized so that the preschool had at least one screen available (1) or preschool had no screens (0).

##### Social Environmental Factors

The early educators were asked to complete a questionnaire related to their own practices, attitudes, and norms related to children’s health behaviors in preschool. The paper-form questionnaire followed a socioecological framework, and consisted of questions from previously used questionnaires [56,57,58], our formative pilot study [59], and items relevant to the Finnish preschool context. Of the items on this questionnaire, three were used as indicators of social environmental factors in preschool because these items are often mentioned as possible key strategies when aiming to reduce children’s ST [29,30]. The preschool group means were aggregated for each measure. 

Early educators’ self-efficacy for children’s PA consisted of one statement: ‘I can persuade my group’s children to be physically active when they want to play sitting still.’ The answer options were from not at all (1) to very much (5).

Early educators’ practice to break children’s ST consisted of one statement that was ‘I have a habit to plan activities during which the children do not have to be still for longer than 30 min’. The answer options were from totally disagree (1) to completely agree (5).

Early educators’ practice of being active with children consisted of one statement: ‘I have a habit to be active together with the children when I am out with them.’ The answer options were from not at all (1) to very much (5). 

##### Organizational Environmental Factors

The principals of the preschools answered a questionnaire in electronic form. The items in the questionnaire were mostly related to rules and regulations related to children’s health behaviors in preschool. In addition, one early educator per preschool group completed an additional short paper-based questionnaire related to practices and regulations of children’s health behaviors in their preschool group. Both questionnaires were based on the previously used questions [53,56,58] and items developed suitable for the Finnish preschool context. For this study, we selected three indicators that early educators mentioned as influential factors for children’s PA and ST in preschool setting in focus group interviews that were held before this cross-sectional study [59]. 

Physical education (PE) lessons was a composite score of four statements. One early educator in each group reported how many PE lessons the children in her/his group had per week indoors or outdoors. In addition, early educators reported on how long (in minutes) one PE lesson was, both indoors and outdoors. The number of weekly lessons was multiplied by the length of one lesson separately for lessons indoors or outdoors. After this transformation, these variables were combined into one continuous variable to illustrate the total minutes of PE lessons in the preschool group. 

PA theme weeks in preschool was based on the questionnaire completed by principals. Principals reported how often preschool has had an organized PA theme weeks during the last two years. The answer options were none, once or twice, and more than twice. Due to the distribution of the answers, these options were recoded so that theme weeks more than twice (1) were compared to other responses (0). 

SB theme weeks in preschool consisted of principal-reported information about how often the preschool had organized SB theme weeks (including screen time) during the last two years. The original answer options were none, once or twice, and more than twice. These options were recoded into none (0) to at least once (1). 

#### 2.3.3. Parental SES

Of all the potential SES indicators (e.g., household income, employment status, occupation), the mother’s educational level was used as an indicator of parental SES, because the mother’s educational level was found to be the strongest, most reliable, and most consistent determinant of children’s health behaviors and childhood obesity in previous studies [60,61,62].

The participants were asked to report their highest education in a ready-made six-item list that went from comprehensive schooling to doctoral degree. To unify the concepts used in this article, we refer to children with higher parental SES (‘children of mothers with higher educational level’) and children with lower parental SES (‘children of mothers with lower educational level’) throughout the remainder of this paper. 

#### 2.3.4. Covariates and Clusters

Children’s age and gender and study season were used as covariates in the analyses. Children’s age was treated as a continuous variable. The study season measure was divided into three categories: 1 = September–October, 2 = November–December, and 3 = January–April. Clustering of children within the preschools and preschool groups was taken into account by conducting multilevel analyses as described below. 

### 2.4. Statistical Analyses

The descriptive statistics, the Spearman correlations, and multicollinearity were checked using the SPSS statistical program version 23 (SPSS Inc., Chicago, IL, USA). Multicollinearity was tested using tolerance and variance inflation factors. Results indicated no issues with multicollinearity [63]. 

Multilevel linear regression models with cross-level interactions were applied to examine the associations between preschool environmental factors and parental SES with children’s ST in preschool. All analyses were adjusted for the child’s age and gender and study season and run by using the Mplus program version 7.13 (Muthen and Muthen, 2016). Multilevel models are the appropriate statistical method when it is necessary to study group-level effects and their interaction with individual level variables in data sets in which persons are nested in groups, such as children attending the same preschool [64,65]. Children were designated as the first level unit, and preschools or preschool groups (depending on the measurement level of the environmental factor) as the second level unit. Each individual level independent variable (child’s age and gender and parental SES) was group-mean centered as recommended when the aim is to study cross-level interactions [65,66,67]. MLR (maximum likelihood with robust standard errors) was used as an estimator in the analyses.

The multilevel analyses were conducted in multiple steps [68,69]. First, the main effects were examined using random intercept and fixed slope models. Each environmental factor was entered separately into the model that also included parental SES as an independent variable. Second, the cross-level interactions were studied by using random intercept and random slope models. In these analyses, we created a slope variable of the association between parental SES and children’s preschool ST and entered each environmental factor separately into the model to explain the variance in this slope. Using this approach in Mplus allowed us to test whether the strength and direction of parental SES–children’s ST association varied according to the preschool environmental factors. Third, to interpret the significant cross-level interactions, we estimated the slope (i.e., the association between parental SES and children’s ST) at different values of the preschool environmental factors. For dichotomous environmental factors, the low and high values were coded as zero and one. For continuous factors, the low, middle, and high values were represented by the minus and plus one standard deviation from the sample mean.

## 3. Results

Participating children were on average 4 years and 4 months old (standard deviation 10 months). Over 70% of the participating children attended preschool at least 8 h per day, and at least 4 days per week. The 821 children wore the accelerometer during measurement week. Of these, 779 children (371 girls, 47.6%) had valid preschool-hour data to be used in the analyses. The average daily wear time during the preschool time was 419 min (standard deviation 56 min). There were no statistically significant differences in gender, age, or SES between the children who had valid data during preschool hours and those who did not. The majority of mothers (41%, *n* = 358) had a bachelor’s degree, whereas 26% (*n* = 226) had a master’s degree. In addition, 18% of mothers (*n* = 150) had a vocational school degree, 3% (*n* = 22) had a comprehensive school background, 9% (*n* = 79) high school, and 3% (*n* = 24) had a licentiate or doctoral degree. The descriptive statistics, the intraclass correlations, and the Spearman correlations of the items used in this study can be found in Table 1. Early educators’ practice of breaking children’s ST often and physical education lessons correlated negatively with children’s ST, whereas early educators’ self-efficacy for children’s PA correlated positively with children’s ST. 

### 3.1. The Associations between Preschool Environmental Factors and Children’s ST

Table 2 presents the associations of preschool environmental factors and the children’s ST adjusted for parental SES. Higher early educators’ practice to break children’s ST was associated with lower levels of children’s ST. Children in preschools with more often conducted PA theme weeks had less ST. Children attending preschools with a higher number of PE lessons had lower levels of ST.

### 3.2. The Moderating Effects of Preschool Environmental Factors on the Associations between Parental SES and Children’s ST

Table 3 presents the effect of each preschool environmental factor on the slope of the association between parental SES and children’s ST. Of all the tested associations, the following preschool environmental factors had a significant effect on the slope: PA theme weeks in preschool, SB theme weeks in preschool, and PE lessons. The moderating effects are presented graphically in Figure 2, Figure 3 and Figure 4. Higher parental SES was associated with higher children’s ST in preschools with organized SB theme weeks, while parental SES and children’s ST were unrelated in preschools without such SB theme weeks. Higher parental SES was associated with higher children’s ST in preschools with a lower amount of PA theme weeks, while parental SES and children’s ST were unrelated in preschools with higher numbers of PA theme weeks. Higher parental SES was associated with higher children’s ST in preschools with a low number of PE lessons, while parental SES and children’s ST were unrelated in preschools with high numbers of PE lessons.

## 4. Discussion

The purpose of this study was twofold: (1) To examine the associations of physical, social, and organizational environmental factors in a preschool setting on children’s objectively measured ST during preschool hours, and (2) to study if these factors in a preschool setting moderate the association between parental SES and children’s ST. The primary findings of this study were that early educators’ practice of breaking children’s ST often, more frequent PA theme weeks in preschool, and longer PE lessons were associated with lower amounts of children’s ST during preschool hours. In addition, we identified three preschool environmental factors acting as moderators in the associations between SES and children’s ST. Higher parental SES was associated with higher children’s ST in preschools (a) with organized SB theme weeks, (b) with lower numbers of PA theme weeks, and (c) with low number of PE lessons.

Our study brought novel knowledge, especially about the associations of social and organizational environmental factors on children’s ST in a relatively large sample, including multiple preschools in different environmental contexts. In contrast to many other studies [29], we did not find significant associations between physical environmental factors and children’s ST. The variation in these results may be due to using composite scores for measuring environmental factors, variations in cultural contexts within which the preschools operate, or the use of different study procedures (observations vs. educator-reported questions).

In a preschool setting, children may have limited choices for movement due to all the inner-built norms, practices, and policies reinforcing a sedentary lifestyle (e.g., early educator-led activities, limited spaces indoors, practices to ask to sit, and norms that require sitting). Sedentariness in these settings is usually an unconscious decision without active consideration because strong cues from the social and physical environment encourage ST [26,70]. In these settings, it may therefore be valuable to pay attention to the amount of ST but also on promoting short bouts of ST [71]. Based on our previous formative study, educators recognized multiple types of practices in the daily preschool structure (e.g., other children sit and wait when outdoor clothes are worn for one child, children sit when morning circles are held) that mainly increased children’s ST. They also acknowledged that due to these practices, some children may have much sedentariness during preschool hours if active attention is not paid to the issue [59]. Similarly, other studies have stated that educators may not understand the role they can play in helping children to be physically active and less sedentary [72,73,74]. In the current study, we recognized that higher early educators’ practice of breaking children’s ST was associated with lower children’s ST. Other studies have suggested that education about ST for educators [75], increasing educators’ motivation to reduce ST [76], or increasing educators’ awareness of children’s prolonged ST in preschool [73] are potentially important strategies in interventions or public health programs in preschools. All these strategies may lead to early educators developing the practice of breaking children’s ST because it may be necessary to have knowledge about the harmful effects of prolonged ST in order to understand the importance of breaking ST. It should, however, be acknowledged that in these settings, it may be an easier target to shift children’s behavior from sitting to standing or increasing light PA instead of moderate-to-vigorous PA [26].

Our findings related to the theme weeks were mixed. More often conducted PA theme weeks were associated with lower children’s ST. Higher parental SES was associated with children’s ST in preschools with organized SB theme weeks or in preschools with a lower number of PA theme weeks. The role of SES may need to be considered more profoundly but carefully when designing the contents of theme weeks. It is unknown what the actual content of these theme weeks has been and how much the role of SES is taken into account when designing their content. Future studies should explore whether our results regarding these interaction effects can be replicated. However, conducting either theme week, PA or SB, may be an indicator of how much certain preschool values exist to educate children about healthy lifestyle habits. The current national curriculum for preschool education, which came into effect after the data collection for this study, underlines that health behavior education should be a part of yearly programs in Finnish preschools. [77].

Another finding was related to PE lessons in preschool. Having higher numbers of PE lessons per week was associated with lower levels of children’s ST. In addition, higher parental SES was associated with higher children’s ST in preschools with a low number of PE lessons. Other studies have also found that children have less ST on preschool days when a PE lesson is included in the daily schedule [78] or if the preschools have regular early educator-led structured activity [79]. Our study brings additional knowledge about the SES differences in the role of PE lessons. As Figure 4 suggests, the number of PE lessons seemed to have minimal influence on ST among children with a low parental SES background. That is, the number of PE lessons in preschool especially influences ST among children from high SES backgrounds. The weekly agenda in preschool usually follows similar patterns. Instead of PE lessons, these preschools with fewer PE lessons may have unstructured free play times. Therefore, children with high parental SES backgrounds may concentrate more on sitting-based activities during unstructured free play times (such as reading books, making puzzles) in these preschools offering fewer PE lessons. Other studies have shown that children with higher parental SES backgrounds may concentrate and pay attention to cognitive tasks longer than children with lower SES backgrounds [80,81]. On the other hand, children with lower SES backgrounds are more restless in the preschool settings, and this may accumulate as children’s higher activity levels [82]. Currently, there are no national guidelines for the weekly number of PE lessons in preschools in Finland [43], but our study suggests that it may be beneficial to use regulations to ensure that all preschools have equal amounts of PE lessons for their children.

To our knowledge, this is the first study to explore the interactions between environmental factors in a preschool setting and parental SES on children’s ST, although there has been a call for studying if environmental factors in certain settings, such as preschool, are associated with SES differences in children’s health behaviors [27]. The interactions we found in our study were small, but consistent in the case of high SES. High parental SES background was associated with higher children’s ST in preschools with organized SB theme weeks, with lower numbers of PA theme weeks, and with low number of PE lessons. These results are thus important in suggesting that the interaction between parental SES and setting may sometimes lead to more sitting and other sedentary behaviors among children with higher SES backgrounds. It may, therefore, be necessary to study further the role of the preschool setting, for instance, if it balances the possible SES differences in children’s health behaviors, because most of the children from different SES backgrounds attend preschool. 

The generalizability of the present findings is limited by the low participation rate of children and it may be that the selected sample of children from certain preschools participated in our study (e.g., the most physically active children or the children with the high SES backgrounds). However, the participation rate of preschools was quite high, allowing us the opportunity to collect data from multiple preschool settings with a variety in physical, social, and organizational environmental factors. We also used a multimethod approach, including observations, self-report questionnaires, and the use of objective and validated measures of ST, although we also relied on self-reported, nonvalidated data for some variables in early educators’ questionnaire. Limitations also include the cross-sectional design limiting conclusions regarding causation. We acknowledge that there are other SES indicators (e.g., income) that may produce different results. The mother’s education as an indicator of SES was chosen due to its highest sample size compared to other SES indicators used in this study. In addition, accelerometers may not be the best possible device to measure ST due to the inability to accurately separate standing from sitting [83]. There is no general agreement on the best possible cut-points, wear-time criteria, and assessment techniques to be used in this age group in accelerometer-based studies [49]. We found three statistically significant interactions out of nine tested associations in our analyses, and it may be that the increased number of statistical tests performed increased the potential for Type I error. Generally, our effect sizes were small, but our results may have multiple practical importance when designing effective preschool interventions aiming to reduce children’s ST.

## 5. Conclusions

Due to the high prevalence of preschool children’s ST in preschool settings, there is a clear need for preschool interventions that reduce ST. To implement effective interventions, those aspects of the preschool setting that have the greatest influence on children’s behavior need to be present. We found that higher early educators’ practice of planning activities that break children’s ST, more often organized PA theme weeks, and longer PE lessons were associated with lower children’s ST. In addition, PA and SB theme weeks, and the minutes of PE lessons per week acted as moderators in the associations between parental SES and children’s ST. Educating early educators to break children’s ST regularly, better-tailored theme weeks according to SES and equal numbers of PE lessons between preschools may be a valuable intervention strategy for reducing children’s ST.

## Figures and Tables

**Figure 1 ijerph-16-00046-f001:**
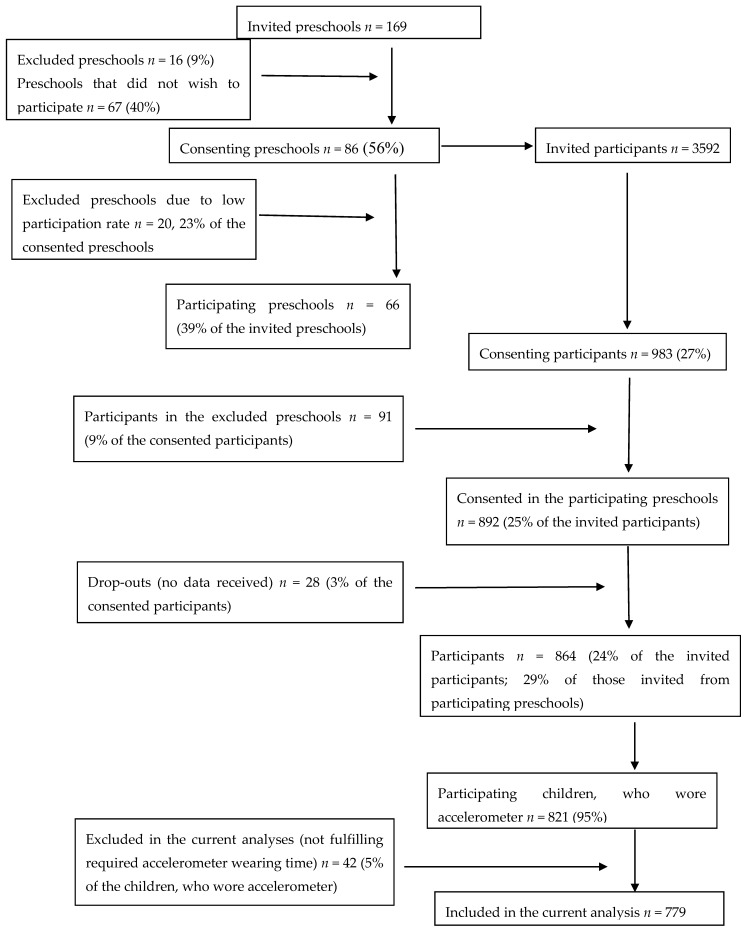
Flowchart of participating preschools and children in the DAGIS (increased health and wellbeing in preschools) cross-sectional study.

**Figure 2 ijerph-16-00046-f002:**
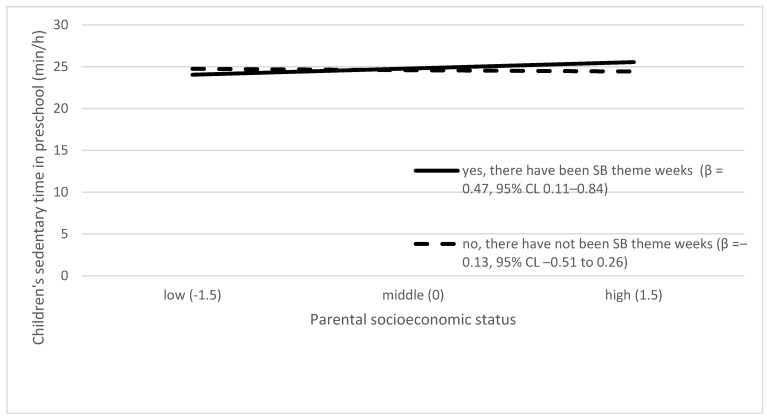
Interaction between SB theme weeks and parental socioeconomic status in predicting children’s preschool sedentary time. Lines in the graphs in Figure 2 represent the dichotomized values yes, there have been sedentary behavior (SB) theme weeks (solid line) and no, there have not been SB theme weeks (dashed line).

**Figure 3 ijerph-16-00046-f003:**
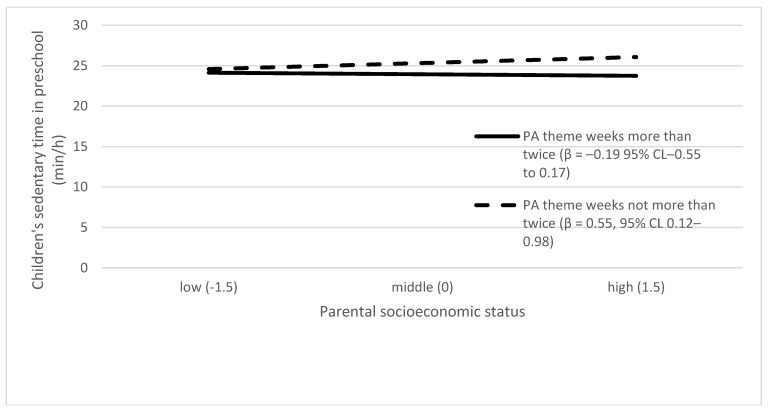
Interaction between PA theme weeks and parental socioeconomic status in predicting children’s preschool sedentary time. Lines in the graphs in Figure 3 represent the dichotomized values there have been physical activity (PA) theme weeks more than twice (solid line) and PA theme weeks not more than twice (dashed line).

**Figure 4 ijerph-16-00046-f004:**
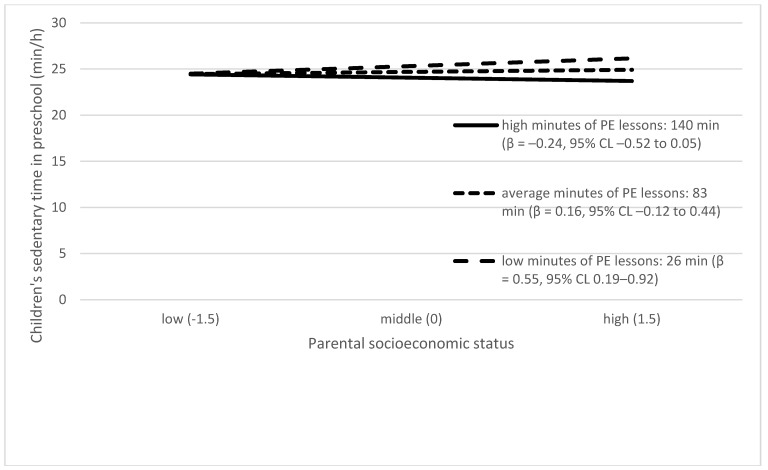
Interaction between PE lessons and parental socioeconomic status in predicting children’s preschool sedentary time. Lines in the graphs in Figure 4 represent the values of 1 standard deviation (SD) above (solid line) and below (dashed line) the mean of physical education PE lessons in preschool (dense dashed line).

**Table 1 ijerph-16-00046-t001:** Descriptives and the Spearman correlations between the measures used in the study (listwise *n* = 577). SES: Socioeconomic status.

Variable	Mean if not Stated Otherwise	Standard Deviation	Intraclass Correlation	1.	2.	3.	4.	5.	6.	7.	8.	9.	10.
1. Children’s sedentary time in preschool (min/hour)	26.47	5.10											
2. Availability of indoor equipment (observed range 1.11–13.56)	7.48	3.16	0.415	−0.037									
3. Availability of outdoor equipment (observed range 9–16)	12.64	1.81	0.413	0.004	0.169 ***								
4. Availability of screens (1 = yes at least one screen, 0 = no at all)	58% had at least one screen		0.418	0.000	0.008	0.047							
5. Early educators’ practice to break children’s ST (scale 1–5)	4.45	0.50	0.413	−0.075 *	0.026	0.100 *	−0.118 **						
6. Early educators’ self-efficacy for children’s PA (scale 1–5)	3.97	0.45	0.412	0.090 *	0.157 ***	−0.001	0.129 ***	0.110 **					
7. Early educators’ practice of being active with children (scale 1–5)	3.74	0.65	0.414	−0.051	0.120 ***	0.080 *	−0.103 **	0.024	0.046				
8. Physical activity theme weeks in preschool (1 = yes, more than twice, 0 = others)	54% more than twice		0.401	−0.066	0.028	0.012	0.067	0.110 *	0.047	0.120 **			
9. Sedentary behavior theme weeks in preschool (1 = at least one time, 0 = others)	38% has had		0.402	−0.025	−0.124 ***	0.147 ***	0.051	−0.061	−0.079 *	−0.097 *	0.302 ***		
10. Physical education lessons (min/week)	82.84	56.93	0.420	−0.083 *	0.083 *	−0.061	0.004	0.041	0.066	0.161 ***	−0.034	−0.136 ***	
11. Parental SES				0.012	0.015	−0.023	0.055	0.030	0.071 *	−0.004	0.029	0.077 *	0.008

* *p* < 0.05, ** *p* < 0.01, *** *p* < 0.001.

**Table 2 ijerph-16-00046-t002:** The associations of parental socioeconomic status and preschool environmental factors with children’s sedentary time.

		β	Lower 95% CI	Upper 95% CI	*p*-Value	Residual Variance %
Physical environment in preschool setting						
Model 1 (*n* = 768)						
	Availability of indoor equipment (level 2)	−0.02	−0.20	0.17	0.855	38
	Parental socioeconomic status (level 1)	0.16	−0.12	0.44	0.265	16
Model 2 (*n* = 771)						
	Availability of outdoor equipment (level 2)	0.08	−0.18	0.33	0.283	38
	Parental socioeconomic status (level 1)	0.15	−0.13	0.44	0.540	15
Model 3 (*n* = 766)						
	Availability of screens (level 2)	0.09	−0.94	1.13	0.860	37
	Parental socioeconomic status (level 1)	0.16	−0.13	0.43	0.278	16
Social environment in preschool setting						
Model 4 (*n* = 750)						
	Early educators’ practice to break children’s sedentary time (level 2)	−1.10	−2.01	−0.11	0.030	41
	Parental socioeconomic status (level 1)	0.16	−0.12	0.44	0.262	15
Model 5 (*n* = 750)						
	Early educators’ efficacy for children’s PA (level 2)	0.54	−0.53	1.62	0.322	39
	Parental socioeconomic status (level 1)	0.16	−0.12	0.44	0.262	15
Model 6 (*n* = 750)						
	Early educators’ practice of being active with children (level 2)	−0.51	−1.30	0.28	0.241	39
	Parental socioeconomic status (level 1)	0.16	−0.12	0.44	0.263	15
Organizational environment in preschool setting						
Model 7 (*n* = 705)						
	Physical activity theme weeks in preschool (level 2)	−1.30	−2.58	−0.02	0.046	41
	Parental socioeconomic status (level 1)	0.12	−0.16	0.40	0.403	15
Model 8 (*n* = 712)						
	Sedentary behavior theme weeks in preschool (level 2)	0.28	−1.18	1.72	0.724	43
	Parental socioeconomic status (level 1)	0.12	−0.16	0.39	0.450	7
Model 9 (*n* = 717)						
	Physical education lessons (min/week) (level 2)	−0.01	−0.02	−0.00	0.035	47
	Parental socioeconomic status (level 1)	0.15	−0.15	0.43	0.327	7

Dependent variable: Children’s sedentary time (min/hour) in preschool (level 1). Indicator of parental socioeconomic status: Mother’s educational level; estimates from multilevel linear regression (random intercept and fixed slope) models. The models were adjusted for children’s age and gender, and study season. The models were clustered by preschool or preschool group. Residual variance: In each model, the first number illustrates level 2 (between preschool or preschool group) residual variance and the second number Level 1 (within preschool or preschool group) residual variance.

**Table 3 ijerph-16-00046-t003:** Effects of preschool environmental factors on the slope of socioeconomic status–children’s sedentary time association.

		Estimate	Lower 95% CI	Upper 95% CI
Physical environment in preschool setting				
Model 1 (*n* = 768)				
	Variation between preschool groups in the slopes (σ)	0.40	−0.38	1.18
	Effect of availability of indoor equipment in preschool on the slope (β)	−0.05	−0.14	0.04
Model 2 (*n* = 771)				
	Variation between preschool groups in the slopes (σ)	0.45	−0.32	1.21
	Effect of availability of outdoor equipment in preschool on the slope (β)	0.07	−0.06	0.20
Model 3 (*n* = 766)				
	Variation between preschool groups in the slopes (σ)	0.42	−0.31	1.16
	Effect of availability of screens in preschool on the slope (β)	−0.28	−0.87	0.31
Social environment in preschool setting				
Model 4 (*n* = 750)				
	Variation between preschool groups in the slopes (σ)	0.42	−0.36	1.19
	Effect of early educators’ practice to break children’s sedentary time on the slope (β)	−0.17	−0.74	0.41
Model 5 (*n* = 750)				
	Variation between preschool groups in the slopes (σ)	0.36	−0.42	1.15
	Effect of early educators’ efficacy for children’s physical activity on the slope (β)	−0.36	−0.97	0.25
Model 6 (*n* = 750)				
	Variation between preschool groups in the slopes (σ)	0.44	−0.33	1.21
	Effect of Early educators’ practice of being active with children on the slope (β)	−0.47	−1.26	0.32
Organizational environment in preschool setting				
Model 7 (*n* = 705)				
	Variation between preschools in the slopes (σ)	0.24	−0.46	0.93
	Effect of physical activity theme weeks in preschool on the slope (β)	−0.63	−1.22	−0.03
Model 8 (*n* = 712)				
	Variation between preschools in the slopes (σ)	0.16	0.16	0.16
	Effect of sedentary behavior theme weeks in preschool on the slope (β)	0.61	0.09	1.12
Model 9 (*n* = 717)				
	Variation between preschool groups in the slopes (σ)	0.19	−0.57	0.94
	Effect of number of physical education lessons (min/week) on the slope (β)	−0.01	−0.01	−0.00

Dependent variable: The slope of the association between parental socioeconomic status (= mother’s educational level) and children’s preschool sedentary time (min/hour). Estimates from multilevel linear regression (random intercept and random slope) models with cross-level interaction. The models were adjusted for children’s age and gender, and study season. The models were clustered by preschool or preschool group.

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
