# Peer review of "Preschool Environmental Factors, Parental Socioeconomic Status, and Children’s Sedentary Time: An Examination of Cross-Level Interactions"

_ijerph, 2018, doi:10.3390/ijerph16010046_

Round 1
Reviewer 1 Report
Abstract
Could you include the method used to measure ST, prior to stating accelerometer in line 22? (I am aware this may not be possible due to wordcount)
Line 24 - I am unsure what you mean by "higher early educators' practice of breaking children's ST" - could this be more clearly stated. It makes more sense when you mention it in your conclusion (line 406)
Introduction
Clear justification for the study. An interesting read - thank you
Material & Methods
Line 87 - is [37] in the right place? Should it be before the fullstop?
Detailed sample description, although I wonder if this would be better displayed as a flow chart
Line 120 - how were possible naps determined? Did you have a valid wear time to be included in the analysis? i.e, min of 10hours?
Availability of screens - this is more a general comment but did you distinguish between screen types and type of behaviour (i..e TV = sitting, tablet = might be sitting or moving)
Line 161 - 'practices' has been used twice, is this an error?
Line 18- - who was the contact person?
Results
You have asked the children to wear the accelerometer for 24hours, yet in the results you seem to only focus on the preschool-hour data, therefore why ask the children to wear the device for 24 hours?
Table 1 - check the formatting (i.e. Standard Deviation). A more detailed description of the results in table 1 may be beneficial
Discussion
Line 325 - check reference placement
Line 327 - could you give examples of types of practices?
Line 395 - you have raised a good point about accelerometers (sitting and standing) so why did you use this tool? Could you support your choice a little further here.
This was a really interesting read and your work adds to the preschool literature with some novel findings. It will be interesting to see how/if your findings are applied in practice regarding PE time, theme weeks and planning breaks.
Author Response
Dear reviewer,
Thank you for the comments and additional questions. We appreciate the comments and questions because these changes in manuscript improve the quality of our manuscript. Hopefully, our changes in the manuscript will further clarify the manuscript. More detailed answers are written below. The changes made in the manuscript are written in red. The possible removals are marked with strikethrough and red colour.
Abstract
Could you include the method used to measure ST, prior to stating accelerometer in line 22? (I am aware this may not be possible due to wordcount) Thank you for this comment. We have modified the abstract so that the method used to measure ST is stated clearly. (please, see the changes in lines 21-23)
Line 24 - I am unsure what you mean by "higher early educators' practice of breaking children's ST" - could this be more clearly stated. It makes more sense when you mention it in your conclusion (line 406) Thank you for this comment. We acknowledge that the word ‘higher’ is not suitable for use in this part. We have now modified this section so that it is clearer to understand. (please, see the changes in lines 26-27).
Introduction
Clear justification for the study. An interesting read - thank you We appreciate this comment.
Material & Methods
Line 87 - is [37] in the right place? Should it be before the fullstop? We have corrected the reference before the full stop.
Detailed sample description, although I wonder if this would be better displayed as a flow chart. It is true that a flow chart may describe better the sample. We have now added a new figure to illustrate the flow of participating preschools and children. Please, see the figure 1 and a sentence in line 125.
Line 120 - how were possible naps determined? Did you have a valid wear time to be included in the analysis? i.e, min of 10hours? We do not have information about children’s nap times during preschool hours. The practices related to nap times in preschools varies, and there is no written policies for that. When planning the data collection methods, we decided not to collect the information of nap times during preschool hours because it would have meant that early educator should have conduct reporting individually for each participating child adding to the overall workload. Therefore, we were not able to take nap times into account when analyzing this data. We are not sure if we understood right the second question. When we did our accelerometer variables, we decided to create five different variables. One of these variables was overall time that had criteria of 600 minutes of wearing time per each included day. However, when creating the preschool hours measure, we decided to set the limit on 4 hours for each included day. This information is provided in the manuscript on lines 187-188.
Availability of screens - this is more a general comment but did you distinguish between screen types and type of behaviour (i..e TV = sitting, tablet = might be sitting or moving). Thank you for this question. We did not distinguish between screen types and type of behavior. We only observed if the preschools had certain type of screens or not. It is however important point to be considered in future studies, and create better instruments to measure how the screens are used in preschools.
Line 161 - 'practices' has been used twice, is this an error? Yes, this is an error. We have removed the second ‘practice’ word.
Line 18- - who was the contact person? Each preschool group could decide independently who was the contact person in their group. We hoped that the contact person was one of the early educators in the preschool group who had worked there for some time and knew the practices in the group. However, we acknowledge that the word ‘contact person’ to be used in this sentence is misleading and not suitable. We have now replaced this word with ‘ early educator’. We have also modified sentences so that the reader understands that this so called contact person questionnaire was different than that of early educators’ questionnaire. Please, see the changes in lines 244-246.
Results
You have asked the children to wear the accelerometer for 24hours, yet in the results you seem to only focus on the preschool-hour data, therefore why ask the children to wear the device for 24 hours? Thank you for this comment. As I mentioned earlier, we have created five different variables of the accelerometer data. These variables have been used in other manuscripts of this DAGIS study. For the purposes of this study, we considered that preschool hours is the best variable.
Table 1 - check the formatting (i.e. Standard Deviation). We have checked the formatting. A more detailed description of the results in table 1 may be beneficial. We have also added detailed description of the results in Table 1 into manuscript (please, see the lines 320-322).
Discussion
Line 325 - check reference placement. We have corrected the reference placement.
Line 327 - could you give examples of types of practices? We have added some examples (please, see the additions in lines 428-429).
Line 395 - you have raised a good point about accelerometers (sitting and standing) so why did you use this tool? Could you support your choice a little further here. When we were designing the data collection methods, we considered multiple options. We wanted to have easy-to-use method to measure children’s movement behaviors. As parents and early educators had already quite many questionnaires and diaries to complete, we decided to have objective measurement for movement behaviors. We also considered the activpal as an option, but we got some feedback from other research groups that children of this age (3-6) have had problems to wear it (e.g. easy to drop down from tight to ankle). In addition, back then in year 2014 and 2015, there was no good cut-points for wrist-worn accelerometer for this age group.
This was a really interesting read and your work adds to the preschool literature with some novel findings. It will be interesting to see how/if your findings are applied in practice regarding PE time, theme weeks and planning breaks. Thank you for this comment.
Reviewer 2 Report
General Comments
Thank you for the opportunity to review this manuscript. It is an interesting topic that I believe is worthy of further exploration and discussion. That said, there are some areas of the manuscript that require revision and strengthening – mainly around the rationale and discussion of implications. I have provided some comments that I believe will help strengthen this paper and its contributions.
I believe there are some important key papers missing from this paper (intro and discussion sections) – namely, work by Tucker et al. (2016 – EPAO and ST in childcare) and Carson and Kuzik (2016 – ST in childcare).
To date, there has been much research examining the impact of the childcare environment (physical and social) on the activity and sedentary behaviours of young children. Besides the inclusion of examining SES, I believe the rationale for this paper could be strengthened. Why is this work novel? What gaps will it be filling? Why is maternal/family SES an important moderator to examine within childcare? – I think this point could be stronger in the introduction.
For SES, why would only maternal education be considered or used as a proxy? Why not employment status and annual income? I think this is a limitation of the study (presents half a picture).
Could the authors clarify why PA and ST theme weeks were examined? Are these typically deemed influential within the childcare literature? Could additional detail be provided to explain the characteristics of these weeks?
To clarify, am I correct in understanding that the accelerometry data was collected raw and then reintegrated into 15s epochs? How did the authors define a valid day of wear time? How many hours of wear time equated to a valid day?
It is understood that the well-cited tool EPAO tool was used to examined particular sedentary attributes within the childcare environment. I’m curious to understand why the document review portion also would not have been employed to provide even deeper insights into the policy/curriculum/practices of sedentary time in childcare. Please explain.
Overall, I think the discussion section requires some work to really drive home key findings/contributions of this research. What were the main implications of this work (namely around the SES work)? Was anything surprising around the SES findings? How does this compare to other published literature? What are some important next steps of natural extensions of this work?
For international audiences, I would urge authors to use consistent terminology throughout. For example, “preschooler” vs. “preschool children” vs. “preschool-aged children” AND “childcare” vs. “preschool” vs. “preschool-type setting”, etc.
I strongly encourage the authors to take a solid review of the manuscript to enhance the English quality of the paper. This will help enhance the readability of the paper overall.
Specific Comments
Line 32 (introduction) – remove “the” after “in”
Line 42 (introduction) – should it actually be that SCREEN-VIEWING (not ST) should be limited to one hour a day (as per guidelines)?
Line 45 (introduction) – should be “aimed at shifting...”
Line 87 (methods) – what is meant by “subjective right”?
Line 120 (methods) – should be “wear time”
Line 243 (results) – add “the” before “accelerometer”
Line 325 (discussion) – delete “the” after “encourage”
For Table 2 – is it possible to provide the % variance accounted for by each model?
Could explanatory titles be added to the 3 figures (rather than just a, b, c)?
Throughout, you can probably present ages as numerical values (e.g., see section 2.1 of paper)
Respectfully submitted.
Author Response
General Comments
Thank you for the opportunity to review this manuscript. It is an interesting topic that I believe is worthy of further exploration and discussion. That said, there are some areas of the manuscript that require revision and strengthening – mainly around the rationale and discussion of implications. I have provided some comments that I believe will help strengthen this paper and its contributions. Dear reviewer,
Thank you for the comments and additional questions. We appreciate the comments and questions because these changes in manuscript improve the quality of our manuscript. Hopefully, our changes in the manuscript will further clarify the manuscript. More detailed answers are written below. The changes made in the manuscript are written in red. The possible removals are marked with strikethrough and red colour.
I believe there are some important key papers missing from this paper (intro and discussion sections) – namely, work by Tucker et al. (2016 – EPAO and ST in childcare) and Carson and Kuzik (2016 – ST in childcare). Thank you for this comment. We have paid attention to these manuscripts, and added their main conclusions into the manuscript (either in introduction or in discussion).
To date, there has been much research examining the impact of the childcare environment (physical and social) on the activity and sedentary behaviours of young children. Besides the inclusion of examining SES, I believe the rationale for this paper could be strengthened. Why is this work novel? What gaps will it be filling? Why is maternal/family SES an important moderator to examine within childcare? – I think this point could be stronger in the introduction. Thank you for this comment. It is true that there are multiple studies on preschool environment and children’s movement behaviors. We have now strengthen our rationale so that the uniqueness of our purposes are clearer and stressed the research gaps that still exist. Please, see the changes in lines 81-89.
For SES, why would only maternal education be considered or used as a proxy? Why not employment status and annual income? I think this is a limitation of the study (presents half a picture). We have added some justification for selecting the maternal education as our proxy (please, see the lines 260 -263). We have also acknowledged in the discussion that other SES indicators may produce different results (line 494).
Could the authors clarify why PA and ST theme weeks were examined? Are these typically deemed influential within the childcare literature? Could additional detail be provided to explain the characteristics of these weeks? Thank you for these questions. In last decade, there has been multiple projects and programs in Finnish preschools aiming to increase children’s PA and decrease ST. Municipality or individual preschool has decided to implement these programs. Some projects have also been nationwide, but the participation for these nationwide campaigns is decided by the preschool. Based on this variety in programs, it is difficult to illustrate what the actual content of each theme weeks has been. Before this cross-sectional study, we conducted some focus group interviews, and in these interviews, early educators stated that there has been plenty of PA theme weeks and some SB/ST theme weeks in their preschools, and these theme weeks have given them some practical tips how to increase children’s PA and decrease ST. However, only small portion of early educators participated in these focus groups, so that we could not estimate how often the theme weeks are actually held. Therefore, we decided to include these questions in the principals’ questionnaire in the cross-sectional study. We believe that this question measures also that how much certain preschool values having topics of health behaviors in their yearly programs. If there are conducted more often theme weeks about these topics, the more the preschool considers them important. We acknowledge that this item is not previously measured, but based on the formative work in the Finnish context, we consider this as relevant topic to measure also in cross-sectional study. Anyhow, we have decided to clarify our selection of these items in the manuscript (please, see the changes in lines 244-246).
To clarify, am I correct in understanding that the accelerometry data was collected raw and then reintegrated into 15s epochs? Yes, this is right. How did the authors define a valid day of wear time? How many hours of wear time equated to a valid day? When we did our accelerometer variables, we decided to create five different variables. One of these variables was overall time that had criteria of 600 minutes of wearing time per each included day. However, when creating the preschool hours measure, we decided to set the limit on 4 hours for each included day. The children attend either half-day (4 hours) or full-day (around 8 hours) in preschool in Finland and therefore we decided to set the time limit in four hours. In our study, the average daily wear time during the preschool time was 419 minutes (standard deviation 56 minutes). However, it should be acknowledged that most of the participating children in our study spent full day in preschool that is 82% of children were at least four days a week in the preschool, and 89 % of the children were at least 7 hours per day in preschool. These information are provided in multiple parts of the original manuscript (please, see the lines 185-188 and 303-306).
It is understood that the well-cited tool EPAO tool was used to examined particular sedentary attributes within the childcare environment. I’m curious to understand why the document review portion also would not have been employed to provide even deeper insights into the policy/curriculum/practices of sedentary time in childcare. Please explain. Thank you for this question. Our observation sheet was partly based on the EPAO tool but we had some additional items from other instruments. When we designed our study and its data collection methods, we also conducted pilot testing of EPAO tool in the Finnish preschools. This pilot testing showed us that the document review part in the EPAO is not suitable for Finnish context. There were multiple items that were not relevant to measure in our context. Especially, many of the sedentary attributes were not relevant because most of the preschools in Finland does not have any screen time activities included in the daily agenda. In addition, we decided to focus on physical environment factors in our observation sheet. So, basically, we have only equipment-related questions from EPAO in our own observation sheet. We have however asked some policy issues from the director and early educator in our questionnaires. The aim is to use these policy items is in other manuscripts.
Overall, I think the discussion section requires some work to really drive home key findings/contributions of this research. What were the main implications of this work (namely around the SES work)? Was anything surprising around the SES findings? How does this compare to other published literature? What are some important next steps of natural extensions of this work? Thank you for this comment. We have now added some new parts to our discussion so that the key messages are acknowledged. Please, see the changes in lines 413-419, 441-450). Due to this new chapter, we modified partly the end of the discussion so that it is more fluent to read.
For international audiences, I would urge authors to use consistent terminology throughout. For example, “preschooler” vs. “preschool children” vs. “preschool-aged children” AND “childcare” vs. “preschool” vs. “preschool-type setting”, etc. We have double-checked our manuscript, and we use ‘preschool children’ and ‘preschool’ now throughout the manuscript.
I strongly encourage the authors to take a solid review of the manuscript to enhance the English quality of the paper. This will help enhance the readability of the paper overall. We have reviewed our manuscript in order to enhance the English quality of the manuscript.
Specific Comments
Line 32 (introduction) – remove “the” after “in” OK.
Line 42 (introduction) – should it actually be that SCREEN-VIEWING (not ST) should be limited to one hour a day (as per guidelines)? No, it is actually ST. At least the Finnish guidelines recommends limiting prolonged sedentary time for one hour and breaking ST often.
Line 45 (introduction) – should be “aimed at shifting...” OK.
Line 87 (methods) – what is meant by “subjective right”? We have removed this word from the manuscript. We used this word to illustrate that it is child’s individual right for preschool place, not for instance families’ right.
Line 120 (methods) – should be “wear time” OK.
Line 243 (results) – add “the” before “accelerometer” OK.
Line 325 (discussion) – delete “the” after “encourage” OK.
For Table 2 – is it possible to provide the % variance accounted for by each model? Thank you for this comment. We are not sure if we have understood right this question, but we have calculated residual variances for each model and added them in the Table 2. We followed the guidance provided by Stride, C. Multilevel Modelling using Mplus. .; Falcon Training/Figure It Out: London, 2013.
Could explanatory titles be added to the 3 figures (rather than just a, b, c)? We have added these explanatory titles in the figures.
Throughout, you can probably present ages as numerical values (e.g., see section 2.1 of paper) we have corrected this so that all the ages are presented as numerical values.
Respectfully submitted.
Reviewer 3 Report
This manuscript describes the moderating effects of preschool environmental factors on the association between parental socioeconomic status (SES; which was in fact maternal education level) and children's sedentary time. Misleading is that children's sedentary time is restricted to preschool hours. In my opinion, this limits the significance of this manuscript to a large extent: it is not clear to me how parental SES would influence children's sedentary time during preschool hours, as children behave according to the routine in the preschool setting. If the author's do have a clear hypothesis on this, it is not clearly mentioned. In fact, the restriction to preschool hours was not mentioned until the methods section (line 123). Moreover, the author's did not give any indication of the variance in parental SES, with 41% of mothers having a bachelor's degree and 26% a master's degree, I doubt whether a sample children with high-educated mothers was included.
Other important limitations include:
- The association between maternal education and children's sedentary time during preschool hours was not provided;
- Table 1 includes intraclass correlation coefficients, does this reflect the inter-obeserver reliability?
- Accelerometer data analysis: non-wear time definition was set at 10 minutes of consecutive zeros, do the authors have a reference for this? Nap times were not excluded; only two days were included.
- Multilevel analyses, adjusting for clustering of children within the preschools: I'm wondering whether the variance in preschool setting is already taken into account by the preschool setting factors?
- Preschool factors: equipment (indoor/outdoor/availability screens) was coded into available or not, but if the children were not allowed to play with, it is actually not available for them.
Author Response
This manuscript describes the moderating effects of preschool environmental factors on the association between parental socioeconomic status (SES; which was in fact maternal education level) and children's sedentary time.Thank you for the comments and additional questions. We appreciate the comments and questions because these changes in manuscript improve the quality of our manuscript. Hopefully, our changes in the manuscript will further clarify the manuscript. More detailed answers are written below. The changes made in the manuscript are written in red. The possible removals are marked with strikethrough and red colour. Misleading is that children's sedentary time is restricted to preschool hours. In my opinion, this limits the significance of this manuscript to a large extent: it is not clear to me how parental SES would influence children's sedentary time during preschool hours, as children behave according to the routine in the preschool setting. If the author's do have a clear hypothesis on this, it is not clearly mentioned. We hope that we have now clarified our purposes in the introduction, please see the changes in lines 73-89, and 95-96). In fact, the restriction to preschool hours was not mentioned until the methods section (line 123). We have now added information about ST during preschools hours already in introduction (please, see the changes in lines 73-89 and 95-96). Moreover, the author's did not give any indication of the variance in parental SES, with 41% of mothers having a bachelor's degree and 26% a master's degree, I doubt whether a sample children with high-educated mothers was included. The distribution of our maternal educational level has been provided in lines 308-311. It is true that we may have sample of higher educated mothers in our study, and it may influence on the results. We have discussed about this (please, see the lines 499-502).
Other important limitations include:
- The association between maternal education and children's sedentary time during preschool hours was not provided; We have added to correlations of maternal education into Table 1.
- Table 1 includes intraclass correlation coefficients, does this reflect the inter-obeserver reliability? The intraclass correlation reflects how strongly certain units in the same group resemble each other and it is assessed to see if clustering is needed in the analyses.
- Accelerometer data analysis: non-wear time definition was set at 10 minutes of consecutive zeros, do the authors have a reference for this? Nap times were not excluded; only two days were included. Thank you for this comment. We have added reference for 10 minutes of consecutive zeros. To be included in the analysis, we required that child was at least two days in the preschool during measurement week. However, most of the children with valid accelerometer data had more than two days.
- Multilevel analyses, adjusting for clustering of children within the preschools: I'm wondering whether the variance in preschool setting is already taken into account by the preschool setting factors? Thank you for this question. We consider that the variance in preschool setting is not taken into account without clustering our analyses. The number of children participating in each preschool group/preschool varies, and therefore, without clustering it may influence the results.
- Preschool factors: equipment (indoor/outdoor/availability screens) was coded into available or not, but if the children were not allowed to play with, it is actually not available for them. It is true what you write. We have not measured if children were allowed to play with the equipment. However, we assume that if there is a variety of equipment available in the preschool, the better possibilities there are also for children to use them. We also consider that equipment in the preschool are meant for children to be used so that early educators also set them to be easily available for children. However, it is definitely an issue to ask in future studies if the equipment available are also allowed to be used by children.
Round 2
Reviewer 2 Report
I believe the authors have done an adequate job at addressing my concerns and appreciate the care taken to strengthen the overall quality of the manuscript.
Two minor points remain:
-I'm still not 100% satisfied with the explanation of how the accelerometry data was handled/processed. I think this needs to be clearer to ensure replicability.
-Once all edits have been made, I would encourage the authors to once again take a final read through the paper to ensure all final typos and grammatical errors have been addressed.
Good luck!